# Fragile X Messenger Ribonucleoprotein Protein and Its Multifunctionality: From Cytosol to Nucleolus and Back

**DOI:** 10.3390/biom14040399

**Published:** 2024-03-26

**Authors:** Mohamed S. Taha, Mohammad Reza Ahmadian

**Affiliations:** 1Institute of Biochemistry and Molecular Biology II, Medical Faculty, Heinrich Heine University Düsseldorf, 40225 Düsseldorf, Germany; mohamed.taha@hhu.de; 2Research on Children with Special Needs Department, Institute of Medical Research and Clinical Studies, National Research Centre, Cairo 12622, Egypt

**Keywords:** FMRP, RNA binding, fragile X messenger ribonucleoprotein protein, protein interaction network, stress granule

## Abstract

Silencing of the fragile X messenger ribonucleoprotein 1 (*FMR1*) gene and a consequent lack of FMR protein (FMRP) synthesis are associated with fragile X syndrome, one of the most common inherited intellectual disabilities. FMRP is a multifunctional protein that is involved in many cellular functions in almost all subcellular compartments under both normal and cellular stress conditions in neuronal and non-neuronal cell types. This is achieved through its trafficking signals, nuclear localization signal (NLS), nuclear export signal (NES), and nucleolar localization signal (NoLS), as well as its RNA and protein binding domains, and it is modulated by various post-translational modifications such as phosphorylation, ubiquitination, sumoylation, and methylation. This review summarizes the recent advances in understanding the interaction networks of FMRP with a special focus on FMRP stress-related functions, including stress granule formation, mitochondrion and endoplasmic reticulum plasticity, ribosome biogenesis, cell cycle control, and DNA damage response.

## 1. General Introduction

A genetic deficiency of the fragile X messenger ribonucleoprotein protein (FMRP, also known as FRAXA, MGC87458, POF, and POF1) results in the most common inherited form of intellectual disability, fragile X syndrome (FXS, also known as Escalante syndrome or Martin–Bell syndrome) [1]. FMRP plays critical roles in germline development during oogenesis [2]; spermatogenesis [3]; the regulation of heart rate during development [4], endothelial cell proliferation, and angiogenesis [5]; stem cell maintenance and differentiation [6]; and tumor progression. FXS patients seem to have a lower risk of developing cancer [7]. Given these numerous, varied, and seemingly fundamental, functions, it is appropriate to recognize the common roles of FMRP throughout the body, i.e., beyond the brain and spinal cord.

FMRP is a well-studied RNA-binding protein (RBP) that regulates local translation [8,9,10,11,12,13,14] and controls calcium channels [15], actin cytoskeletal dynamics [16,17,18], chromatin dynamics [19], DNA damage response (DDR) [19,20], and replication stress response [21]. These cellular functions presume physical properties for FMRP, which are required for both the recognition and localization of messenger RNA (mRNA) targets and the direct association with a multitude of proteins and protein complexes [22,23]. FMRP consists of an N-terminal domain comprising two Tudor (Tud) domains and one K homology 0 (KH0) domain, a central region comprising two KH1 and KH2 domains, and a C-terminal domain comprising a phosphorylation site [24] and an arginine–glycine–glycine (RGG) region [25]. FMRP displays a nuclear localization signal (NLS), a nuclear export signal (NES), and two nucleolar localization signals (NoLSs) [23,26,27,28,29], consequently localizing FMRP to different subcellular compartments in the cytosol and nucleus (Figure 1) [23]. Nuclear FMRP has been suggested to regulate the DDR and genomic stability as a chromatin-binding protein [19].

The most prominent and studied function of FMRP is translational regulation. Moreover, FMRP is a member of the FXR protein family that includes fragile X-related proteins 1 (FXR1P) and 2 (FXR2P). They share high sequence conservation with their N-terminal and central regions and form heteromeric complexes [30,31,32,33]. All three FMRP family members are RNA-binding proteins that regulate the translation of their cargo mRNAs and associate both physically and functionally with the microRNA pathway [34]. FMRP has been shown to suppress the translation of its target mRNAs via association with either stalled, non-translating polyribosomes or microRNA [35,36,37,38]. This can lead to the formation of cytoplasmic ribonucleoprotein (RNP) granules, which control the expression, repression, or decay of specific mRNAs [39]. There are many types of cytoplasmic RNA granules, which differ markedly in size, composition, and mode of biogenesis [40]. Some granules, such as processing bodies (P-bodies) and stress granules (SGs), which are assembled by many cell types, transport, store, or degrade mRNAs, thereby indirectly regulating protein synthesis [39,41,42,43]. Accordingly, they contribute to various aspects of cellular homeostasis. There is increasing evidence suggesting that such RNP granules are associated with several age-related neurodegenerative diseases [44]. However, the molecular networks that regulate and guide FMRP towards these cellular processes need further investigation.

## 2. Integrated Stress Response

The cell responds to various stressors by adapting its cellular functions and biological processes. This results in a wide range of dynamic and morphological changes affecting all subcellular structures, such as nucleoli, mitochondria, and the cytoplasm. The prompt integrated cellular response to stress is crucial to avoid pathophysiological consequences [45]. FMRP is associated with several kinases, including protein kinase RNA-activated (PKR) and phosphatidylinositol 4-kinase alpha (PI4KA), as well as a wide range of stress-related proteins, such as RAS-GTP-activating protein SH3 domain-binding protein (G3BP), ubiquitin-associated protein 2-like (UBAP2L), Ataxin-2-Like Protein (ATXN2L), cell cycle-associated protein 1 (CAPRIN1), FXR1P, FXR2P, DEAD-Box Helicase 3 X-Linked (DDX3X), and Nuclear FMRP Interacting Protein 2 (NUFIP2) (Figure 1 SG) [46,47,48,49]. This suggests that FMRP acts as a stress sensor for different types of stressors, such as viral infection, cellular toxicity, and DNA damage [47,50]. Consequently, other working groups have anticipated the role of FMRP in stress-related pathways, such as cell apoptosis and tissue tumor necrosis [51,52]. The integrated stress response typically begins with the halt of cellular translation and the formation of SGs [53]. This function requires continuous protein trafficking between the cytoplasm and the nucleus, which may be facilitated by the interaction of FMRP with either the importin/karyopherin superfamily member KPNA2 (Importin α1) or nucleoporins proteins (NUPs) [47,54].

## 3. Stress Granules Formation

SGs are reversible and dynamic membrane-less assemblies of proteins, mainly nucleoproteins and RNA, formed in the cytoplasm of eukaryotic cells in response to stress [46]. Recently, SGs have become the focus of translational medicine studies for the treatment of various infectious and non-infectious diseases, including cancer [55,56]. Recent studies have shown that stress granule formation can attenuate the phenotypes of neurodegenerative diseases [57].

A variety of environmental, endogenous, and genetic factors can induce a cellular stress response by forming SGs and inhibiting general protein translation [58]. This response is triggered by the phosphorylation of eukaryotic initiation factor 2α (eIF2α) by PKR [59].

The presence of RNA-binding proteins and RNA is critical for stress granule assembly and liquid–liquid phase separation (LLPS). The intrinsically disordered regions (IDRs) of RNA-binding proteins, such as G3BP1, TIA-1, CAPRIN1, and FMRP, contribute to LLPS and stress granule nucleation [60,61,62]. UBAP2L is an interacting protein of FMRP and is thought to be a core protein in SGs [63,64]. Arginine methylation in the RGG domain of the UBAP2L protein is responsible for the recruitment of G3BP1 to stress granule assembly [65,66]. Interestingly, the interaction between UBAP2L and FMRP was identified in two different proteomic studies. The first study used the amino- and carboxy-terminal domains of FMRP in HeLa cells [47]. The second study used nuclear FMRP isoform 6, which is approximately 86% homologous to FMRP isoform 1, in osteosarcoma cells. Importantly, FMRP isoform 6 has a different carboxyl-terminal domain and lacks the nuclear export signal (NES) present in FMRP isoform 1 [50]. In the mouse brain, approximately 30% of FMRP is associated with ribosomal proteins [43]. However, according to Asano-Inami et al., FMRP is less abundant in SGs compared to other proteins [67]. In particular, it was found to be present at lower levels compared to UBAP2L, Protein Arginine methyltransferase 1 (PRMT1), FXR1P, FXR2P, G3BP1, and G3BP2, respectively. Among these proteins, UBAP2L is the most abundant in SGs [67]. In addition to its role in stress granule formation and translation inhibition, FMRP phosphorylation is critical for translation activation by releasing its binding partner YTH N6-methyladenosine RNA-binding protein F1 (YTHDF1) [68]. This suggests that FMRP can positively and/or negatively regulate different cellular processes. Taken together, these findings suggest that FMRP may have a potential role both during and after the cessation of stress.

## 4. Mitochondrion and Endoplasmic Reticulum Plasticity

FMRP regulates calcium homeostasis by regulating the contact point between the ER and mitochondria [69]. In early 2014, we noticed a remarkable association of FMRP with the mitochondria, which colocalize with the mitochondrially encoded cytochrome C oxidase II (MTCO2) [23]. Other proteomic studies, including ours, have supported the role of FMRP in mitochondrial homeostasis. This was achieved by analyzing the proteomic changes in mitochondria in the presence and absence of FMRP [70], or by identifying its association with mitochondrial proteins [47].

Mitochondrial dysfunction and oxidative stress are common phenomena in almost all neurodegenerative diseases [71,72]. Other possible roles of FMRP may be in mitophagy and mitochondrial biogenesis through its interactions with sequestosome 1 (SQSTM1) and the ATPase family, AAA domain-containing protein3 (ATAD3) [47,73,74]. SQSTM1 is recruited to putative mitochondrial serine/threonine kinase (PINK1)-induced mitochondrial clusters and may be involved in the clearance of mitochondria [75,76]. SQSTM1 not only assists and links PINK1 to the mitophagy pathway but is also a selective cargo receptor for autophagy to degenerate misfolded proteins [73].

FMRP is associated with the ATPase family, AAA domain-containing protein 3B (ATAD3B), which plays a role in mitochondrial biogenesis (Figure 1 ME) [77]. ATAD3A and ATAD3B are also involved in ER–mitochondrial interaction [74]. Complement C1q-binding protein (C1QBP), another FMRP-interacting protein appears to be critical for mitochondrial energy metabolism [78]. Its deficiency has been suggested to be a major cause of leukoencephalopathy due to axonal degeneration [79].

FMRP interaction with single-stranded DNA-binding protein (SSBP1) may support cell survival and the maintenance of the mitochondrial membrane potential against proteotoxic stress by forming a complex with the heat shock factor HSF1 and facilitating the recruitment of the chromatin remodeling factor BRG1 [80]. The role of FMRP in controlling cytoskeletal dynamics and microtubule formation may indirectly affect mitochondrial mobility [81], as well as mitochondrial functions and morphology in Drosophila [82]. In mice, ER stress induces FMRP phosphorylation through the activation of inositol-requiring enzyme-1 (IRE1), which, in turn, reduces atherosclerosis [83]. Recently, Phosphatidylinositol 4-kinase III α (PI4KA) has been identified as an interacting partner with the nuclear isoform of FMRP (isoform 6) [50]. PI4K produces phosphatidylinositol 4-phosphate (PI4P), which is then converted to Phosphatidylinositol 4,5-bisphosphate (PIP2). The latter serves as a regulator for ion channels and transporters and interacts with the proteins involved in endo-/exo-cytosis regulation. Finally, phosphatidylinositol 3,4,5-trisphosphate (PIP3) and diacylglycerol are generated from PIP2 and phospholipase C by PI3K [84]. PIP3 controls various processes such as phagocytosis, exocytosis, and cytoskeletal organization [85]. However, further research is needed to better understand the role of ER and inositol-3-phosphate (IP_3_) receptors in neurodegeneration.

## 5. Ribosome Biogenesis

Nucleoli are membrane-less subcellular organelles that are morphologically and dynamically affected by stress. They are considered to be the center of the cellular stress response [86,87]. The complex and dynamic assembly and maturation of the eukaryotic ribosome, which is called ribosome biogenesis, is orchestrated by ribosomal proteins, approximately 500 rRNAs, small nucleolar RNAs (snoRNAs), and trans-acting factors [88]. FMRP localizes to the nucleolus [23], where it may participate in the biogenesis of ribosomal subunits, given the large number of FMRP-associated proteins, which are involved in various stages of ribosome biogenesis [47].

Nucleophosmin (NPM1) participates in several stages of ribosome biogenesis, initiated by rRNA transcription. NPM1 binds at G-quadruplex regions, at ribosomal DNA, gene promoters, and also the c-MYC promoter, which is a regulator of rRNA processing and maturation [89,90]. Cyclin-dependent kinase inhibitor 2A (CDKN2A) inhibits ribosomal biogenesis through its physical interaction with nucleolar phosphoprotein NPM1, RNA helicase DDX5, and RNA polymerase I termination factor TTF-I (Figure 1 RB) [91]. CARF, as a collaborator of ARF, regulates the early steps of pre-rRNA processing during ribosome biogenesis by controlling the spatial distribution of XRN2, a 5′-3′ exoribonuclease, between the nucleoplasm and nucleolus [92]. Other FMRP-associating proteins involved in pre-rRNA processing are FtsJ RNA 2′-O-methyltransferase 3 (FTSJ3), MYB-binding protein 1a (MYBBP1A), N-acetyltransferase 10 (NAT10), Nucleolar Protein 2 (NOP2), PUF (Pumilio/FBF)-A, and TSR1 [93,94,95].

NAT10, for example, is a lysine acetyltransferase and targets not only microtubules and histones but also the 30S precursor of 18S rRNA [96]. Active nuclear protein complexes of PKR and NPM1 were recently described to be associated with ribosome biogenesis [97]. On one hand, ribosomal RNA processing 12 homologs (RRP12) and the exportin CRM1 are involved in the late assembly of the 40S ribosomal subunit in the nucleolus [98], and LTV1 is involved in the nuclear export of the 40S ribosomal subunit [99]. TSR1 acts as a quality control checkpoint in the maturation of 40S ribosomal subunits [100]. On the other hand, nucleolar eIF6, SDAD1 (SDA1 Domain Containing 1), and SKI2 are components of the pre-ribosomal particles. NFAR and eIF6 are required for the assembly and maturation of 60S subunits, respectively [101,102]. The GTPase NOG2 binds to immature nucleoplasmic 60S particles and blocks the association of the nuclear export adapter protein NMD3 [103]. Cytosolic GTPase LSG1, along with RPL10, is required for releasing the 60S subunits from nuclear export adapter NMD3 in the cytoplasm [104].

## 6. Cell Cycle Control

A series of events, such as cell growth, DNA replication, and mitosis, take place during the cell division cycle, which underlies the cell-cycle checkpoint activation, caused, for example, by DNA damage and replication stress [105]. A large number of FMRP-interacting proteins are involved in the control and modulation of the cell cycle [47]. GPC1 regulates the cell cycle and proliferation by suppressing cell cycle inhibitors, including p21WAF1 (CDKN1A), p27 (CDKN1B), p16INK4a (CDKN2A), p19 (CDKN2D), and D-type cyclins, together with inducing CDK2 and SKP2 [106]. More specifically, CDKN2B and CDKN2D contribute to G2/M cell cycle arrest [107], while CDK2 promotes G1/S transition [108]. The arginine methyltransferase PRMT5 binds to CDK4 and activates CDK4-RB-E2F-mediated transcription by releasing CDKN2A from CDK4 [109]. Interestingly, SND1 binds and activates E2F and also modulates G1/S transition [110]. CARF differently regulates the cell cycle by undergoing a complex crosstalk with other proteins including p16INK4a, Retinoblastoma protein (pRB), Human double minute 2 protein (HDM2), and p21WAF1 (CDKN1A) [111]. CARF is also crucial for DNA damage and the checkpoint response of cells through ATM/CHK1/CHK2, p53, and ERK pathways [112]. On the other hand, pathways like ATM/CHK2 and CHK2/ERF regulate G2/M progression through regulating CDC25C [113]. In response to genotoxic agents, CDC5L interacts with ATR and is required for the S-phase cell-cycle checkpoint [114]. RPL6 appears to bind to HDM2 and attenuates HDM2-mediated p53 ubiquitination and degradation [115]. EWSR1 regulates mitosis by dynamically influencing microtubule acetylation [116], while TPX2 is a microtubule-associated protein that is required for mitotic spindle function [117]. FAU, a ubiquitin-like protein, covalently coupled to BCL-G, a pro-apoptotic member of the B-cell lymphoma 2 (BCL-2) family, regulates UV-induced apoptosis [118]. Lastly, FAM120A regulates the activity of SRC kinases to protect cells from oxidative stress-induced apoptosis [119]. The patient-derived FXS astrocyte model showed altered cell cycle dynamics, characterized by shortened S-phase length and increased expression of cyclin D1, a regulator of the G1/S checkpoint [120]. In mouse brain neurons, FMRP interferes with cell cycle regulation through its interaction with cell division cycle 20 (CDC20). CDC20 is responsible for ubiquitinating and downregulating the anaphase-promoting complex/cyclosome (APC/C), and FMRP activates the E3 ubiquitination (Figure 1 CC) [121]. In the nuclear fraction of rat forebrains, FMRP was found to be associated with several nuclear pore complex proteins (NUP153, NUP37, and NUP93), where it was suggested that FMRP controls the function of the nuclear pore complex (NPC) proteins during mitosis [54,122].

## 7. DNA Damage Response

The DDR is a network of cellular pathways that sense, signal, and repair DNA lesions. These pathways strictly control the cell cycle checkpoint, chromatin remodeling and transcriptional programs, DNA repair, and, in the case of severe damage, apoptosis. Different studies have provided the first insights into the role of FMRP in DDR [19,21,123]. It controls, in complex with staufen double-stranded RNA-binding protein 1 (STAU1) and TAR DNA-binding protein 43 (TDP-43), the expression and synthesis of SIRT1, a deacetylase involved in the repair of DNA double-strand breaks (DSBs) [123]. The induction of DNA damage by DNA polymerase inhibitors alters the nuclear FMRP interaction protein network through the recruitment of DNA damage response proteins such as aurora kinase B (AURBK), DNA topoisomerase II alpha (TOP2A), heterochromatin protein 1 binding protein 3 (HP1BP3), kinesin family member 22 (KIF22), and epithelial cell transforming 2 (ECT2) [50]. It has been reported that nuclear isoforms of FMRP regulate or inhibit the formation of DNA bridges through direct binding during mitotic anaphase. Depletion of these isoforms leads to the accumulation of DNA bridges [124].

On the one hand, it has been implicated that FMRP is involved in a feed-forward mechanism that triggers a replication stress-induced DDR [21]. On the other hand, Alpatov et al. have previously demonstrated that FMRP participates in the DDR in a chromatin-binding-dependent manner [19]. They have shown that FMRP binds chromatin through its tandem Tud domain in vitro and associates with chromatin in vivo. Thus, FMRP is recruited to chromatin in response to replication stress [19], and Drosophila FMRP in the nucleus is involved in a replication stress-induced DDR [21]. FMRP is thought to regulate genomic stability at the chromatin interface by negatively modulating CHD1 function. The FMRP- and PAF1 complex (PAF1C-) associated tumor suppressor CHD1 represents an ATP-dependent chromatin remodeling protein [47]. Deletion of CHD1 is most common in prostate cancer, which is characterized by genomic instability [125]. Another proteomic study identified chromodomain helicase DNA-binding protein 4 (CHD4) as an FMRP interactor in rat neuronal cells. CHD4 regulates DNA repair and chromatin remodeling via its interaction with zinc finger protein (ZNF) 410 or via the formation of nucleosome remodeling and the deacetylase (NuRD) complex with Ewing sarcoma RNA-binding protein (EWSR1) [126,127,128].

High levels of FMRP were identified in human cancer [129]. The embryonic lethality of BRCA1/2 mutations has been discussed to be rescued by the presence of low *FMR1* alleles that are characterized by less than 26 CGG repeats [130]. In this regard, there is a functional relationship between FMRP and BRCA1. On the one hand, BRCA1 cooperates with the positive transcription elongation factor b (P-TEFb) and the FMRP-associated NUFIP to activate transcription through RNAP II [131]. On the other hand, BRCA1 regulates the biogenesis of noncoding microRNAs by forming a protein interaction network with the DROSHA microprocessor complex, SMAD3, p53, and the FMRP-associating RNA helicases DDX5 and DHX9 [132]. Remarkably, DDX5 undergoes interactions with other different proteins. It is critical for the p53-induced expression of p21WAF1/CIP1, leading to cell-cycle arrest in response to DNA damage [133]. It interacts with c-MYC and modulates its transcription and transforming activity [134]. Strikingly, p14Arf blocks the physical interaction between DDX5 and c-MYC [134]. Another FMRP-interacting protein that regulates p53 is ubiquitin-specific peptidase 10 (USP10). An early DNA damage response is ATM-mediated stabilization and the nuclear translocation of USP10, which, in turn, activates p53 [64]. SSBP1 recruits the checkpoint complex to initiate ATR signaling [135]. Collectively, FMRP appears to be involved in mechanisms underlying growth-inhibitory effects, given the interdependence with diverse tumor suppressors.

The FMRP-associating proteins Exosome Component 10 protein (EXOSC10), Heterogeneous nuclear ribonucleoprotein U (hnRNPU), MATR3, Non-POU domain-containing octamer-binding protein (NONO), poly (ADP-ribose) polymerase 1 (PARP1), PTB-associated splicing factor (PSF), tripartite motif containing 28 (TRIM28), Vigilin, and X-ray repair cross-complementing 6 (XRCC6) play critical roles in response to DDBs. A striking event is the recruitment of NONO, probably in complex with PSF and MATR3 [136], to DNA damage sites, which depends on poly-ADP-ribosylation, via PARP1 [137]. NONO and TRIM28, in turn, promote non-homologous end joining (NHEJ) and attenuate homologous recombination (HR) (Figure 1 DDR) [137]. Thereby, BCL2-associated transcription factor 1 (BCLAF1) stabilizes the Vigilin/XRCC6/DNA-PKcs complex and facilitates NHEJ-based DSB repair in the surviving cells [138,139], while exosome component 10 protein (EXOSC10) is required for the recruitment of RAD51 to DSBs [140]. Interestingly, DNA end resection inhibits NHEJ and triggers homology-directed DSB repair [141]. hnRNPU regulates DNA-end resection by binding to the DSB sensor complex MRE11-RAD50-NBS1 (MRN) [63], a process that appears to be modulated by SQSTM1 [142]. The FMRP-associating, deubiquitinating enzyme Valosin-containing protein (VCP) is involved in orchestrating the proper association of P53-binding Protein 1 (53BP1), BRCA1, and RAD51, three factors critical for DNA repair and genome surveillance mechanisms [143].

## 8. Conclusions and Future Perspectives

The wealth of published data on FMRP has significantly deepened our understanding of its biological functions, greatly expanded the physical and functional protein and RNA interaction networks of FMRP, and suggested its involvement in various fundamental cellular processes throughout the body beyond the central and peripheral nervous systems. Accordingly, FMRP functions may begin in the nucleolus after cytoplasmic-nuclear translocation, where it may be involved in ribosomal subunit biogenesis and, most likely, nuclear export. FMRP may be part of the transcription factory, regulating gene expression through interaction and orchestration of RNA polymerase II, where it binds directly to a large number of mRNAs and transports them to sites of local translation. Upon any type of cellular stress, FMRP accumulates at sites of stress response and facilitates, for example, the stabilization of double-stranded RNA binding and the activation of PKR, leading to the formation of stress granules. In addition, recent FMRP interactome analysis suggests that FMRP may play central roles in DNA damage response, cell cycle regulation, intracellular trafficking, and actin dynamics and that FMRP virally induces innate immune responses. Of note, FMRP may also be involved in mitochondrial quality control and mitophagy, functions directly linked to neurodegenerative and cognitive disorders, including FXS, Huntington’s disease, Alzheimer’s disease, Down syndrome, and progressive supranuclear palsy.

Many research groups have expressed a growing interest in the analysis of FMRP interacting partners. This is mainly due to the limited success achieved so far in understanding the function of FMRP through its full-length purification and determination of its complete structure. In this review, we have focused on the FMRP interaction network as a key factor in understanding its multifunctional properties. The FMRP interaction network sheds light on the intricate structural domains and post-translational modifications of FMRP. FMRP is involved in diverse and highly interconnected cellular processes. For example, its role in cell cycle control is based on checkpoints that depend on DNA integrity and its role in the DNA damage response. Its role in many different diseases, such as cancer, viral infection, innate immunity, and neurodegeneration, is expected to depend on its stress-related function. Focusing on FMRP’s role in stress will be a possible hallmark for understanding the association between elevated oxidative stress and different groups of diseases (e.g., diabetes, cancer, neurodegeneration, and even aging) and consequently in different types of cells.

Given neuronal cells’ limited differentiation and renewability, they must have a robust anti-stress mechanism to counteract their increased vulnerability. Any potential breakdown in this mechanism can lead to the formation of pathological aggregates and subsequent neuronal dysfunction. Therefore, gaining a comprehensive understanding of the role of FMRP in the stress response will serve as a fundamental pillar in the prevention and treatment of various diseases associated with pathological aggregation. In addition, the absence of FMRP reflects the impact of increased stress on multiple processes and cellular organelles, particularly mitochondrial energy and calcium homeostasis.

## Figures and Tables

**Figure 1 biomolecules-14-00399-f001:**
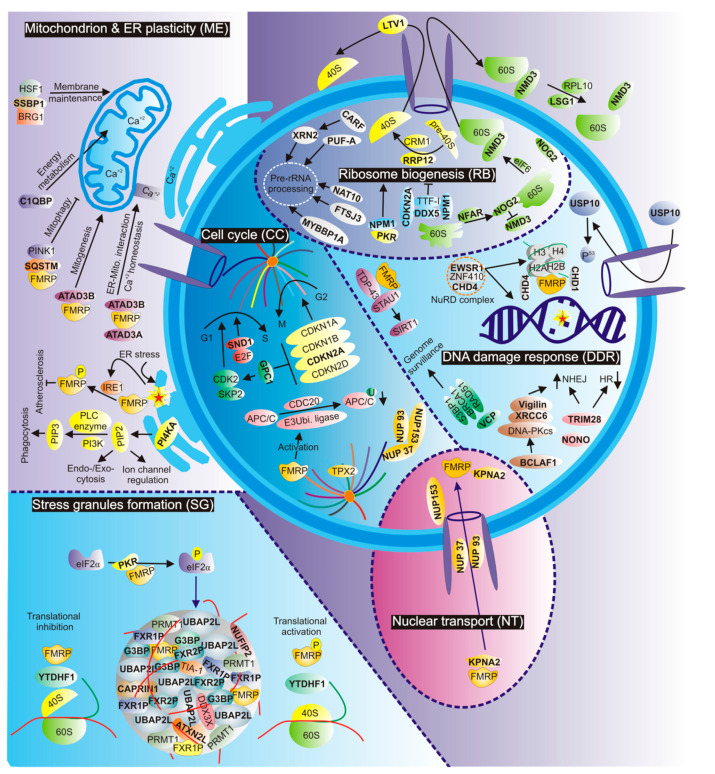
Stress-related functions and interaction networks of FMRP. Proteins in bold have been identified as direct or indirect FMRP interactors. Important FMRP functions related to stress granule formation (SG), mitochondrial and endoplasmic reticulum plasticity (ME), ribosome biogenesis (RB), cell cycle (CC), and DNA damage response (DDR) are highlighted. Red and green lines in (SG) represent RNA and protein, respectively. For further details, see the text.

## Data Availability

It is affirmed that no new data were generated while compiling this review manuscript. All referenced data sources are openly accessible and appropriately cited within the manuscript. Please do not hesitate to contact the corresponding author if any additional information or clarification is required.

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
