# Peer review of "Fragile X Messenger Ribonucleoprotein Protein and Its Multifunctionality: From Cytosol to Nucleolus and Back"

_biomolecules, 2024, doi:10.3390/biom14040399_

Round 1

Reviewer 1 Report

Comments and Suggestions for Authors

This paper is a deep and complex review of all of the functions and interactions of FMRP. I found it enlightening but it is intense in scope. The figure is excellent and very detailed. The English was fine and I only found one error, the spelling of FMRP in line 253 under the section on DNA damage response.  It is worthy of publication.

Author Response

This paper is a deep and complex review of all of the functions and interactions of FMRP. I found it enlightening but it is intense in scope. The figure is excellent and very detailed. The English was fine and I only found one error, the spelling of FMRP in line 253 under the section on DNA damage response.  It is worthy of publication.

Response: Thank you for your enthusiastic feedback. It is a pleasure for us to have your approval of our manuscript. The error (line 253) has been corrected (see yellow text).

Reviewer 2 Report

Comments and Suggestions for Authors

Silencing of the fragile X mental retardation 1 (FMR1) gene and consequently lack of syn-

thesis of FMR protein (FMRP) are associated with fragile X syndrome, which is one of the most prevalent inherited intellectual disabilities. FMRP is a multifunctional protein involved in many cellular functions in nearly all subcellular compartments under normal conditions and under conditions of cellular stress in both neuronal and non-neuronal cell types. 

This is achieved through its trafficking signals, nuclear localization signal (NLS), nuclear export signal (NES), and nucleolar localization signal (NoLS), as well as its RNA and protein binding domains, and is modulated by various post-translational modifications such as phosphorylation, ubiquitination, sumoylation, and methylation. 

This review summarizes recent advances in understanding the interaction networks of FMRP with a special focus on FMRP stress-related functions, including stress granule formation, mitochondrion and endoplasmic reticulum plasticity, ribosome biogenesis, cell cycle control, and DNA damage response.

Comments

A well-written, thoughtful, relevant and comprehensive review aimed at gaining a comprehensive understanding of the role of FMRP in the stress response toward serving as an important pillar in the prevention and treatment of various diseases associated with pathological aggregation. 

Very minor suggestions. 

Lines 10-11, abstract

“Silencing of the fragile X mental retardation 1 (FMR1) gene”

Please use new nomenclature, the Fragile X Messenger Ribonucleoprotein 1 (FMR1) gene.

Line 26, “Genetic deficiency of the fragile X mental retardation protein (FMRP;” 

Please use new nomenclature, the Fragile X Messenger Ribonucleoprotein (FMRP).

Author Response

Comments

A well-written, thoughtful, relevant, and comprehensive review aimed at gaining a comprehensive understanding of the role of FMRP in the stress response toward serving as an important pillar in the prevention and treatment of various diseases associated with pathological aggregation.

Response: Thank you for your enthusiastic feedback and valuable comments. All changes are highlighted in yellow.

Very minor suggestions

Lines 10-11, abstract and line 26: “Silencing of the fragile X mental retardation 1 (FMR1) gene”. please use the new nomenclature, the Fragile X Messenger Ribonucleoprotein 1 (FMR1) gene.

Response: We appreciate your thoughtful suggestion to replace the outdated FMRP nomenclature with the new one, which we have gladly followed. Thank you very much.

Reviewer 3 Report

Comments and Suggestions for Authors

The manuscript biomolecules-2907163 entitled FMRP and its Multifunctionality: From Cytosol to Nucleolus and Back by Mohamed S. Taha , Mohammad Reza Ahmadian summarizes recent advances in understanding the interaction networks of FMRP with a special focus on FMRP stress-related functions, including stress granule formation, mitochondrion and endoplasmic reticulum plasticity, ribosome biogenesis, cell cycle control, and DNA damage response.

The review work is well written, systematic and updated.

Figure 1 is of high quality and very informative

Minor revision: a linguistic revision is recommended.

Comments on the Quality of English Language

Minor revision: a linguistic revision is recommended.

Author Response

The review work is well-written, systematic, and updated.

Figure 1 is of high quality and very informative.

Response: Thank you for your enthusiastic feedback. It is a pleasure for us to have your approval of our manuscript. The error (line 253) has been corrected (see yellow text).

Minor revision: a linguistic revision is recommended.

Response: The language of the manuscript has been carefully checked.